# A Machine Learning Approach Unveils the Relationships between Sickness Behavior and Interoception after Vaccination: Suggestions for Psychometric Indices of Higher Vulnerability

**DOI:** 10.3390/healthcare11222981

**Published:** 2023-11-18

**Authors:** Gaspare Alfì, Graziella Orrù, Danilo Menicucci, Mario Miccoli, Virginia Casigliani, Michele Totaro, Angelo Baggiani, Angelo Gemignani

**Affiliations:** 1Department of Surgical, Medical and Molecular Pathology and Critical Care Medicine, University of Pisa, 56126 Pisa, Italy; g.alfi@phd.unipi.it (G.A.); danilo.menicucci@unipi.it (D.M.); angelo.gemignani@unipi.it (A.G.); 2Department of Clinical and Experimental Medicine, University of Pisa, 56126 Pisa, Italy; mario.miccoli@unipi.it; 3Department of Translational Research and of New Surgical and Medical Technologies, University of Pisa, 56126 Pisa, Italy; v.casigliani@studenti.unipi.it (V.C.); michele.totaro.unipi@hotmail.com (M.T.); angelo.baggiani@unipi.it (A.B.)

**Keywords:** COVID-19, Sickness Behavior, Interoceptive Awareness

## Abstract

Objective: Prior research has suggested a possible connection between vaccination and manifestations of Sickness Behavior; however, a need remains to first delve deeper into this association and second examine how Interoceptive Awareness and emotional factors may modulate individuals’ perceptions of their health status post vaccination. Method: An online retrospective cross-sectional survey of 647 individuals who received a COVID-19 vaccination was conducted. Together with vaccination side effects, socio-demographic characteristics, health status, level of concern about vaccination, and Interoceptive Awareness were collected at the baseline level. Mood, sleep, and Sickness Behavior were assessed at baseline and after vaccination. Data were analyzed using inferential statistics and machine learning techniques. Results: After vaccination, there was a significant increase in Sickness Behavior levels (mean (±SD) SicknessQ T0 = 1.57 (±2.72), mean (±SD) SicknessQ T1 = 5.54 (±5.51); *p*-value = 0.001; ES = 0.77). A Machine Learning analysis revealed specific patterns of individual dispositions (sex and age), baseline emotional characteristics (levels of depression, anxiety, stress, and concern about adverse reactions), as well as some components of Interoceptive Awareness (Noticing, Body Listening, and Attention Regulation), as predictors of high levels of Sickness Behavior, both in terms of overall scores (JRIP: 72.65% accuracy, AUC = 0.692, d = 0.709; F1 = 0.726) and individual items (JRIP: 75.77% accuracy, AUC = 0.694; d = 0.717; F1 = 0.754). Conclusions: Our results provide new insight into post-immune reactions by highlighting the contribution of Interoceptive Awareness in modulating the severity of Sickness Behavior. This sheds light on the role of awareness of bodily sensations in modulating perceptions of health status, helping to identify the characteristics that make individuals more prone to feeling sick.

## 1. Introduction

Coronavirus disease 2019 (COVID-19) has affected over seven hundred million people worldwide, causing almost seven million deaths (https://covid19.who.int, accessed on 25 September 2023, at 15.16 p.m.) since it was declared a pandemic by the World Health Organization (WHO) on 11 March 2020. The development of vaccines has been conducted at an unprecedented pace to alleviate the resulting burden and complications [1].

Although COVID-19 vaccines have significantly mitigated attack rates, hospitalizations, and deaths [2], immunization-related reactogenicity [3] is a very common response: soon after vaccination, people can show physical manifestations of the associated inflammatory response that may both cause local (pain, redness, and swelling of the injection site) and systemic effects (fever, fatigue, headache, and flu-like symptoms).

These symptoms are known to be associated with the Sickness Behavior (SB) condition, a reflex of the immune activation specifically linked to the activity of pro-inflammatory cytokines (Interleukin-1 β, Interleukin-6, and Tumor Necrosis Factor-α) triggered by an infectious agent or an injury [4]. Consequently, the cytokines, via the vagus nerve and along with the activation of gastrointestinal and cardiovascular reflexes, transmit the altered state of the body to the brain, giving rise to the profound behavioral changes that characterize SB [5,6]: fever, fatigue, lethargy, loss of appetite, social withdrawal, malaise, loss of libido, hyperalgesia, sleep disturbances, and cognitive weakness.

Furthermore, SB includes an emotional component as sick individuals usually experience increased levels of anxiety and a dampening of their mood to the point of mimicking depressive behavior [7], suggesting that it may also lead to pathogenic behaviors.

However, the way individuals experience this condition varies. Responses to sickness conditions can differ greatly: while some of us can feel poorly with mild infections, others can feel well even when more serious conditions and significant physical symptoms are present. Thus, the extent of SB appears to be influenced by several factors, including individual (e.g., age, sex, and gender identity) and cultural aspects [8].

Herein, we hypothesize that in addition to the previously mentioned factors, interoceptive awareness (IA), which is how each of us feels and perceives internal bodily sensations and processes, as well as the attention we pay to changes in our bodies [9], may have an impact on the severity of SB.

Indeed, through the interplay between a specific network of cortical brain regions (including the insula and the anterior cingulate cortex), and peripheral signals originating from the vagus nerve or related to molecular changes (e.g., hormones, cytokines), IA plays a crucial role in maintaining homeostasis and in influencing emotional experience [10,11].

Therefore, it simultaneously enables the monitoring of bodily states and the appropriate response to them, as well as tuning bodily sensations to specific emotions, helping people to recognize and understand their own emotional states [7].

Taking advantage of the COVID-19 vaccination campaign, we collected data through an online survey to retrospectively investigate whether vaccine-induced SB could be modulated by Interoceptive Awareness alongside emotional disposition perceived before vaccination (at baseline).

## 2. Materials and Methods

### 2.1. Online Survey

A cross-sectional online survey, aimed at exploring the potential association between individual characteristics and vaccination side effects, was carried out retrospectively between March and June 2021. The survey was conducted anonymously, with a guarantee of strict confidentiality regarding all participant information. Notably, no data pertaining to the IP addresses used for survey participation were collected.

The study was conducted in accordance with the Declaration of Helsinki and was approved by the Ethics Committee of the University of Pisa. In order to take part to the study, all participants provided written informed consent (presented in the first part of the survey and mandatory to proceed with the questionnaire), declared themselves to be native Italian speakers, and were over the age of 18 years. In order to send the questionnaire responses, participants had to completely fill it out.

Each participant received the first dose of the same type of vaccine, and the online survey was made available in a flexible period ranging from one month to three months after the injection.

The survey consisted of three main sections. In the first one (Appendix A), socio-demographic characteristics were self-reported by participants who answered about their sex, age, civil status, education, level of physical activity, and past or current pathologies (including past COVID-19). Next, they were asked whether the vaccine had produced side effects and if so, to specify which ones through multiple choice in a list of possibilities including both local (e.g., pain, warmth, itching or bruising in the site of injection) and systemic (e.g., feeling tired or generally feeling unwell, feeling sick, decreased appetite, joint pain or muscle pain, fatigue) adverse reactions. Also, the level of concern and precautions before vaccination were investigated.

The second section assessed the individual IA using the Multidimensional Assessment of Interoceptive Awareness (MAIA, Cronbach’s α ranges from 0.66 to 0.82 across the subscales) composed of eight components: Noticing, Not—Distracting, Not—Worrying, Attention Regulation, Emotional Awareness, Self—Regulation, Body Listening, and Trusting [12].

The third section pertained to sickness symptoms, quality of sleep, and the mood of participants, referring retrospectively to both before and the week following the first dose of the COVID-19 vaccination. The Sickness Questionnaire (SicknessQ, Cronbach’s α of 0.86) was used to detect the presence and the severity of SB [13]. The Depression Anxiety Stress Scale—Short Form (DASS-21, Cronbach’s α of 0.90) was used to assess negative emotional states of depression (DASS-21 depression, Cronbach’s α of 0.82), anxiety (DASS-21 anxiety, Cronbach’s α of 0.74), and stress (DASS-21 stress, Cronbach’s α of 0.85) [14]. Quality of sleep was assessed using the Pittsburgh Sleep Quality Index (PSQI, Cronbach’s α of 0.835), composed of seven components: subjective quality of sleep, sleep latency, sleep duration, habitual sleep efficacy, sleep disorders, use of hypnotic drugs, and disorders during the day [15].

### 2.2. Data Pre-Processing

The online survey was implemented using Google Forms. Raw data were then preprocessed in order to be suitable for statistical analysis. In particular, the subjects’ health status responses, as they were collected as free text, were managed, and reported diseases were classified into the five classes of diseases according to the International Classification of Diseases 10th revision (ICD-10) [16], with the addition of the category “other diseases” for health conditions not included in the manual. The question about COVID-19 positivity was collected as multiple choice. As a result, the data were converted into categorical variables and analyzed at a dichotomous level: the presence or absence of diseases belonging to each category.

The adverse reactions to the vaccine were originally collected by asking for the presence/absence of each possible one within a prepared list and then clustering the responses into three categorical variables: fever > 38 °C, local symptoms, and systemic symptoms. Consequently, they were analyzed dichotomously for the presence or absence of symptoms within each cluster. Responses related to the concern about an adverse reaction to the vaccine were collected using a 10-point Likert scale and analyzed as continuous variables.

### 2.3. Data Analysis

A repeated-measures ANOVA was performed, and the mean and standard deviation were reported for the following variables, referring to the baseline and post COVID-19 vaccination (T0 and T1, respectively): SicknessQ, DASS-21 Total Score, DASS-21 components (DASS-21 Depression, DASS-21 Anxiety, DASS-21 Stress), and PSQI Total Score.

Additionally, for all variables that achieved statistical significance (except SicknessQ), a repeated-measures ANCOVA was carried out, incorporating the changes from the post COVID-19 vaccination and baseline (delta) values of SicknessQ as covariates. This allowed us to investigate whether the pre–post effect could be explained by the variation in Sickness Behavior.

For each variable, the effect size (d) was derived [17] and its magnitude was interpreted according to Cohen (1998) [18]: 0.0–0.1 (no effect), 0.2–0.4 (small effect), 0.5–0.7 (intermediate effect), and 0.8–≥1 (large effect). The statistical significance level was set at 5%. In order to investigate the independent variables that could possibly correlate with a high level of SB after COVID-19 vaccination, we divided the participants (n = 647) into two subgroups: participants with a lower level of Sickness Behavior (L-SB), as measured by the SicknessQ at T1, and participants with a higher level of SB (H-SB). For the L-BS group, we retained participants within the 33rd percentile of SicknessQ scores at T1 that ranged from 0 to 2 (n = 252) in our sample. Analogously, for the H-SB group, we retained participants above the 66th percentile of the SicknessQ scores at T1 that ranged from 7 to 28 (n = 227) in our sample. The data analysis was carried out using the Statistical Package for Social Sciences (SPSS Inc., Chicago, IL, USA) and Weka 3.9.

With the aim of assessing the features contributing to the classification of participants with lower and higher levels of post-vaccination SB, we applied Machine Learning (ML) techniques and relevant algorithms. In particular, the classification was based on a propositional rule learner, Repeated Incremental Pruning to Produce Error Reduction [19], implemented in Weka software (JRIP). JRIP uses incremental error pruning to produce error reduction, followed by a complex optimization step; it was selected in this study since it has the advantage of performing classification by selecting a subset of features and providing intelligible decision rules based on this selection. Thus, the classification performance of JRIP was compared to the most common classifiers in order to ensure an appropriate approach. For the comparison, the following classifiers were used: Naïve Bayes [20], Logistic Regression [21], Simple Logistic [22], Support Vector Machine [23], and Random Forest [24]. The classifiers used were run using the default parameters of the Weka software.

In order to have a lower rate of bias and avoid overfitting, we used the k-fold cross validation technique, which randomly divides the dataset into k folds of equal size, where the value for k was set to 10 (k = 10). The quality of classification was estimated by means of the AUC-ROC (AUC) and F1-score (F1) metrics: F1 can have a lowest value of 0 and a maximum value of 1. The highest value indicates perfect precision (correct positive predictions relative to all positive predictions) and recall (correct positive predictions relative to all actual positives), and the lowest possible value is 0. With regards to the AUC, when the value is 1, the classifier is able to accurately distinguish between every positive and every negative class point. In contrast, if the AUC had been 0, the classifier would have predicted all positives as negatives and all negatives as positives.

The classification analysis was carried out in consecutive steps that gradually increased the level of specificity of the potential predictors. All ML models were estimated using the following data: (1) five sociodemographic features; and (2) thirteen items of health status information, including physical side effects after vaccination. Additionally, based on the level of analysis, we progressed from studying potential predictors among the questionnaire’s total scores to evaluating specific questionnaire subscores and items contributing to scores that were identified as predictors at the previous analysis level.

For the classification, we only used the DASS-21, PSQI, and MAIA questionnaire scores/subscores, as well as the corresponding items related to the baseline before vaccination, discarding any information from the week following vaccination.

## 3. Results

A total of six hundred and forty-seven Italian-speaking participants (384 men and 263 women; mean age, 49.87 ± 10.56 (mean ± SD), range 20–70 years) completed the online survey. Demographic and clinical characteristics of the participants are reported in Table 1, while Table 2 reports the adverse effects to vaccination (Appendix A, report trait characteristics).

A repeated-measures ANOVA was performed, and the results are shown in Table 3. The difference of the mean between the paired variables at baseline and after COVID-19 vaccination was statistically significant for the following measures: SicknessQ, DASS-21, and all subcomponents. A large effect size was shown for the SicknessQ scores (T0 vs. T1; d = 0.77) (SicknessQ boxplots are reported in the Appendix A), and the smallest effect was found for the DASS-21 stress score (T0 vs. T1; d = 0.22). Including SicknessQ as a covariate in the repeated-measures ANCOVA for the significant variables allowed us to remove the contribution of SicknessQ to the pre–post changes. Of note, the observed changes in anxiety appear to be completely explained by SicknessQ since the main effect disappears when controlling for this variable (Table 4).

### Machine-Learning-Based Classification Accuracies of High and Low Levels of Sickness Behavior Referred after COVID-19 Vaccination

In order to investigate the independent variables that possibly predict a high level of SB after COVID-19 vaccination, as measured by the SicknessQ in the week after the vaccination, the participants’ data were divided into two subgroups: those with a lower level of SB (L-SB) and those with a higher level of SB (H-SB). Sociodemographic information, health status and DASS-21, and the PSQI and MAIA questionnaire scores were used as independent variables for the predictions made using different ML methods.

In the first preliminary round, all baseline variables were used, including the SicknessQ scores. Regardless of the classifier, the classification accuracy in distinguishing the L-SB and H-SB classes was around 77% (Appendix A). In order to exclude possible trait effects on the individual levels of SicknessQ, the classification was then repeated after excluding SicknessQ at baseline from the ML analysis. During the classification process, the variables exhibiting optimal efficiency in classifying the participants within the L-SB and H-SB groups were identified through JRIP’s rules and their cut-offs. JRIP yielded an accuracy of 72.65% (decision matrix (AUC = 0.692, d = 0.709; F1 = 0.726), Appendix A) by identifying the following eight rules:(1)If DASS-21 Stress T0 ≥ 4 and Fever > 38 °C = Positive, then the subject is classified as H-SB;(2)If Systemic Symptoms = Positive and MAIA Noticing ≥ 1.75 and Sex = Female and Age ≤ 50, then the subject is classified as H-SB;(3)If DASS-21 Depression T0 ≥ 1 and Systemic Symptoms = Positive, then the subject is classified as H-SB;(4)If Systemic Symptoms = Positive and MAIA Attention Regulation ≤ 2 and MAIA Noticing ≥ 2.5, then the subject is classified as H-SB;(5)If Systemic Symptoms = Positive and DASS-21 Anxiety T0 ≥ 1, then the subject is classified as H-SB;(6)If Systemic Symptoms = Positive and PSQI Sleep Latency T0 ≥ 1 and MAIA Body Listening ≤ 2.333333 and MAIA Body Listening ≥ 2, then the subject is classified as H-SB;(7)If Systemic Symptoms = Positive and Concerns about adverse reactions ≥ 5 and MAIA Attention Regulation ≤ 2.714286 and Local symptoms = Absent, then the subject is classified as H-SB;(8)If the previous seven rules are not applicable, then the individual is classified as an L-SB subject.

In addition to the JRIP rules, the classification was also performed using five other common classifiers: Naïve Bayes, Logistic Regression, Simple Logistic, Support Vector Machine, and Random Forest. Their classification accuracy was estimated using the k-fold cross validation technique (k = 10), yielding a performance slightly higher than JRIP (Appendix A). 

The most accurate classifier in classifying L-SB individuals (199/252; 79%) and H-SB ones (175/227; 77%) was Simple Logistic (AUC = 0.819, d = 1.289; F1 = 0.78).

As the final step, the classification was repeated, focusing the analysis on the variables that JRIP indicated as exhibiting optimal efficiency in classifying the individuals within the two classes. Taking into account the eight JRIP rules, the variables were (1) DASS-21 subcomponents, Stress, Depression, and Anxiety; (2) fever > 38 °C; (3) systemic symptoms; (4) local symptoms; (5) concerns about adverse reactions; (6) MAIA Noticing, Attention Regulation, and Body Listening; (7) sex; (8) age ≤ 50; (9) PSQI Sleep latency. In this particular instance, with the goal of identifying more specific aspects capable of predicting the SB level after vaccination, the classification was performed using as independent variables the individual items contributing to all the aforementioned variables. Accordingly, we ran the classifiers using the following as inputs: (1) age; (2) sex; (3) fever > 38 °C; (4) systemic symptoms; (5) local symptoms; (6) concerns about adverse reaction; (7) MAIA items from 1 to 4 (MAIA—Noticing); MAIA items from 11 to 17 (MAIA—Attention Regulation), MAIA items from 27 to 29 (MAIA—Body Listening); (8) PSQI items 2 and 5 (sleep latency); and (9) all items of the DASS-21 (DASS-21 Stress, Depression, and Anxiety). In this instance, JRIP yielded an accuracy of 75.37% with a simplified set of five rules:(1)If DASS-21 Item 11 (“I found myself getting agitated”) T0 ≥ 1 and MAIA item 2 (“I notice when I am uncomfortable in my body”) ≥ 2 and DASS-21 Item 13 (“I felt down-hearted and blue”) T0 ≥ 1, then the subject is classified as H-SB;(2)If systemic symptoms are present and DASS-21 Item 12 (“I found it difficult to relax”) T0 ≥ 1, then the subject is classified as H-SB;(3)If systemic symptoms are present and Sex = Female and Age ≤ 50, then the subject is classified as H-SB;(4)If systemic symptoms are present and concerns about adverse reaction ≥ 4 and MAIA Item 11 (“I can pay attention to my breath without being distracted by things happening around me”) ≤ 2, then the subject is classified as H-SB;(5)If the previous four rules are not applicable, then the individual is classified as an L-SB subject.

The confusion matrix (AUC = 0.694, d = 0.717; F1 = 0.754) showed a correct classification of 180 out of 227 H-SB subjects (accuracy: 79%) and 181 out of 252 L-SB individuals (accuracy: 72%). The accuracies demonstrated by the five main classifiers previously used, using 10-fold cross validation, did not differ significantly compared to the previous one (Appendix A).

## 4. Discussion

This study aimed to determine the association between COVID-19 vaccination and SB as well as identify personal characteristics that predict higher SB levels, with a focus on IA. The results revealed a significant increase in SB following vaccine inoculation, accompanied by heightened levels of depression, anxiety, and stress, which are typical symptoms of SB [4]. Indeed, the phenomenon of Sickness Behavior explained part of the exacerbated emotional distress exhibited by patients following vaccination, specifically contributing to a comprehensive understanding of heightened anxiety levels. This insight points out that Sickness Behavior is not necessarily directly associated with a pathological condition. Despite the statistically significant increases in depression, anxiety, and stress levels observed, these elevations do not reach a clinically relevant magnitude. Therefore, it is more reasonable to consider them typical emotional correlates of the body’s response to the immune challenge.

A hierarchical ML analysis identified specific combinations of characteristics associated with higher SB levels, including female sex, younger age, the presence of systemic symptoms, and specific baseline levels of depression, anxiety, stress, sleep latency, and concern about adverse reaction. Furthermore, high SB levels were associated with the absence of local symptoms. These data suggest that the inability to locate the area of pain or discomfort could favor the emergence of a global condition of malaise.

As hypothesized, the role of IA emerged. Three out of eight MAIA components, Noticing, Body Listening, and Attention Regulation, in combination with other factors, were found to contribute to the classifying individuals into the high-SB group. This sheds new light on the role that the awareness of bodily signals may have in modulating perception of health status. Interoception, which involves predicting and controlling bodily signals, plays a vital role in maintaining homeostasis and providing a flexible allostatic response to complex demands [11]. Based on our results, it appears that becoming aware of body discomfort (Noticing and Body Listening) and the ability to sustain and control attention on it at specific levels (Attention Regulation) may increase the possibility of experiencing more severe SB. In particular, the Noticing component refers to abilities to observe with awareness and experience interoceptive stimuli through mind–body listening, regardless of the state of anxiety [12]; we might contend that noticing body discomfort (becoming aware) may increase the possibility of experiencing more severe SB.

The Body Listening component measures the ability to listen to the body, to observe mindfully, and the ability to not have emotional awareness difficulties [12]. It appears that individuals exhibiting systemic symptoms, a prolonged sleep latency (score > 1), together with a specific level of bodily listening (between 2 and 2.3) may be more vulnerable to the perception of high levels of SB.

The Attention Regulation component measures the ability to sustain and control attention on bodily sensations, which is required for accepting them. It enables one to maintain emotional clarity while remaining connected to emotional and sensory experiences [12]. On the contrary, the lack of the ability to mindfully observe bodily sensations could make subjects less able to regulate their own sensations and consequently be more reactive to immune challenges. This effect could be amplified by the co-presence of a higher level of the Noticing component (Rule 4).

In this scenario, possible strategies emerge; an increasing body of evidence indicates that meditation-based practices, such as Mindfulness-Based Stress Reduction, can have positive effects on attention control, emotional regulation, and self-awareness by acting specific neural circuits involved in interoceptive processes [25,26]. These could alleviate the distress of perceiving a high level of SB.

## 5. Limitations

This study has some limitations. Firstly, we encountered challenges in comprehending the reasons behind the altered sleep quality of our sample prior to vaccine administration, as indicated by the results of the PSQI questionnaire. Secondly, although similar, the two groups of patients classified as L-SB and H-SB are not entirely balanced. Thirdly, the methodological approach may be susceptible to recall bias: the memory of the previous emotional state could be influenced by the emotional state experienced at the time of completing the questionnaire. Additionally, it is worth noting that the participants underwent the vaccination from one to three months before responding to the online survey. This can particularly impact the accuracy of pre-vaccination experiences. Fourth, despite identifying some predictors of SB severity, we currently lack an explanation for the underlying mechanisms and their interaction with interoceptive variables.

## 6. Conclusions

Our findings suggest that SB can be partially attributed to an individual’s emotional disposition at baseline and some components of IA.

It is important to emphasize that SB should not be misconstrued as a negative indicator or evidence that vaccines are harmful. Rather, it reflects a normal response of the body to vaccination. Indeed, from a molecular standpoint, SB can be attributed to the effects of inflammatory mediators that, via the vagus nerve, facilitate communication between the immune response and the CNS following vaccine administration [3]. This physiological response is a part of the body’s natural defense mechanisms and should not be misinterpreted as a vaccine-related issue.

Nevertheless, insights gained from the ML analysis indicate that the integration and regulation of bodily sensations have the potential to alter perceptions of one’s health status. This could potentially aid in identifying characteristics that may be associated with a higher risk of experiencing post-immune challenge malaise, offering a basis for providing tailored support and interventions.

## Figures and Tables

**Table 1 healthcare-11-02981-t001:** Demographic and clinical characteristics of participants (n = 647).

**Demographic Characteristics**	
Sex (male)	59.35% (n = 384)
Age	49.87 (±10.568)
Physical activity practitioners	72.64% (n = 470)
**Civil Status**	
Unmarried/nubile	29.67% (n = 192)
Married	60.43% (n = 391)
Separated	8.65% (n = 56)
Widower	1.23% (n = 8)
**Main Diseases During Lifespan**	
Class II (neoplasm)	3.40% (n = 22)
Class IV (endocrine, nutritional, and metabolic diseases)	7.88% (n = 51)
Class IX (diseases of the circulatory System)	5.56% (n = 36)
Class X (diseases of the respiratory System)	2.47% (n = 16)
Class XI (diseases of the digestive System	1.39% (n = 9)
Past COVID-19 positivity	2.93% (n = 19)
Presence of an immunocompromised state	0.15% (n = 1)
Other diseases	5.1% (n = 33)

Note: frequencies expressed in %; mean (±standard deviation).

**Table 2 healthcare-11-02981-t002:** Adverse effects to vaccination (n = 647).

No Reaction	13.60% (n = 88)
**Fever (>38 °C)**	36.93% (n = 239)
**Local Symptoms**	
Swelling or redness where the injection is given	6.49% (n = 42)
Tenderness, pain, warmth, itching, or bruising where the injection was given	40.05% (n = 270)
**Systemic Symptoms**	
Feeling tired (fatigue) or generally feeling unwell	55.17% (n = 357)
Chills or a feeling of fever	40.80% (n = 264)
Tenderness	40.95% (n = 265)
Joint pain or muscle pain	40.49% (n = 262)
Headache	39.10% (n = 253)
Feeling sick (nausea)	10.04% (n = 65)
Malaise (vomiting or diarrhea)	3.86% (n = 25)
Drowsiness or feelings of dizziness	9.42% (n = 61)
Decreased appetite	5.40% (n = 35)
Enlarged lymph nodes	2.00% (n = 13)
Excessive sweating, itching, or rash	2.16% (n = 14)

**Table 3 healthcare-11-02981-t003:** Repeated-measures ANOVA analysis for state variables referring to baseline and post COVID-19 vaccination (T0 and T1, respectively).

State Variables	Mean	Standard Deviation	*p*-Value	Observed Power	Effect Size
SicknessQ T0	1.57	(±2.72)	0.001	1.00	0.77
SicknessQ T1	5.54	(±5.51)			
DASS-21 Depression T0	1.06	(±2.56)	0.001	1.00	0.49
DASS-21 Depression T1	1.32	(±2.74)			
DASS-21 Anxiety T0	0.65	(±1.64)	0.001	1.00	0.48
DASS-21 Anxiety T1	1.11	(±2.22)			
DASS-21 Stress T0	2.12	(±3.27)	0.001	0.872	0.22
DASS-21 Stress T1	2.34	(±3.51)			
DASS-21 Total Score T0	3.83	(±6.61)	0.001	1.00	0.49
DASS-21 Total Score T1	4.77	(±7.53)			
PSQI Total Score T0	6.03	(±3.09)	0.409	0.131	0.017
PSQI Total Score T1	6.08	(±3.25)			

**Table 4 healthcare-11-02981-t004:** Repeated-measures ANCOVA for state variables achieving statistical significance when comparing baseline and post-COVID-19 vaccination results.

State Variables	F	*p*-Value	Observed Power
DASS-21 Depression	10.228	0.001	0.891
DASS-21 Depression * Delta SB	162.184	0.000	1.00
DASS-21 Anxiety	0.362	0.548	0.092
DASS-21 Anxiety * Delta SB	149.847	0.000	1.00
DASS-21 Stress	6.153	0.013	0.697
DASS-21 Stress * Delta SB	62.732	0.000	1.00
DASS-21 Total Score	6.492	0.011	0.720
DASS-21 Total Score * Delta SB	166.087	0.000	1.00

## Data Availability

The data presented in this study are available upon request.

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
