# Peer review of "A Machine Learning Approach Unveils the Relationships between Sickness Behavior and Interoception after Vaccination: Suggestions for Psychometric Indices of Higher Vulnerability"

_healthcare, 2023, doi:10.3390/healthcare11222981_

Round 1

Reviewer 1 Report

Comments and Suggestions for Authors

The useful finding here is that there may be emotional/mental health factors that mitigate the side effects of vaccine that we do not always consider. This point is important. 

However, I can see that the way that this is presented ("SicknessQ") that in the hands of disinformation content creators that this will become proof that the vaccines are harmful when many of the symptoms are normal vaccine side effects. It would be good to highlight the emotional/mental health point even more and to also be proactive in counteracting the potential disinformation. Address the mis/disinformation issue directly -- this study will be deliberately misrepresented regardless what you do, but at least we could make it hard to do so.

I have some question about the split into H-SB and L-SB groups. It does not seem to take into consideration the severity of the symptions. It looks like if someone has a minor side effect like tenderness in the arm that they fall into the H-SB category and thus are treated with strong symptoms. If this is not the case, the construction of H-SB needs clarification. If this is true, this could be addressed in one of two ways: (1) separate analyses of local and systemic symptoms or (2) an extended discussion of the issue as a limitation.

Author Response

Dear esteemed reviewer,

We would like to express our sincere gratitude for your valuable time and expertise in reviewing our manuscript entitled "A Machine Learning approach unveils the relationships between Sickness Behavior and Interoception after vaccination: suggestions for psychometric indices of higher vulnerability". Your insights and recommendations have been immensely helpful in improving the quality and rigor of our research. In response to your constructive feedback, we have made the following modifications to address your concerns and enhance the overall quality of the paper.

______________________________________

Regarding the first part of the feedback (misinformation), we have taken steps to address this matter as per your suggestions, and you can find the pertinent discussion.at page 9 (lines 358-364) as stated below:

“It is important to emphasize that SB should not be misconstrued as a negative indicator or evidence that vaccines are harmful. Rather, it reflects a normal response of the body to vaccination. Indeed, from a molecular standpoint, SB can be attributed to the effects of inflammatory mediators that, via the vagus nerve, facilitate communication between the immune response and the CNS following vaccine administration [8]. This physiological response is a part of the body's natural defense mechanisms and should not be misinterpreted as a vaccine-related issue.”

Regarding the second part of the feedback provided: We appreciated your suggestion and accordingly we have discussed that H-SB appear to be associated only to systemic symptoms after vaccination, while the absence of local symptoms has been paradoxically included in the decision rules to classify people with H-SB. A possible explanation is that the impossibility to locate the area of pain or discomfort could favor the emergence of a global condition of SB, see lines 307-312, at page 8, as stated below:

“The hierarchical analysis of ML identified specific combinations of characteristics associated with higher SB levels, including female sex, younger age, presence of systemic symptoms, and specific baseline levels of depression, anxiety, stress, sleep latency and concern about adverse reaction. Furthermore, high SB levels were associated with the absence of local symptoms. This data suggests that the inability to locate the area of pain or discomfort could favor the emergence of a global condition of malaise.”

All changes have been highlighted in yellow colour.

Regarding SB classification:

However, regarding the SB classification, it is important to note that the concept of ‘’symptom’’ encompasses not only physical aspects but also, emotional, and affective dimension, there may indeed be individuals experiencing relatively mild physical discomfort. Our decision to classify participants into L-SB and H-SB categories was motivated by the need to understand the factors that might intensify this behavior. In doing so, we paid special attention to interoceptive variables, recognizing that SB goes beyond the scope of physical pain.

Reviewer 2 Report

Comments and Suggestions for Authors

dear Authors,

the present text addresses an study to connect vaccination and sickness manifestations to reveal possible emotional factors in the perception of the symptoms. This matter is of great interest due to the social commotion and controversy that exists around the Covid disease and the massive vaccination processes that have occurred.

The methods and apporaches used, by inferential statistics and Machine Learning are giving the possibility of a more in-depth analysis with variables that are somewhat imprecise in their quantification, as are the emotional characters. The results, not being totally conclusive, do provide behavioral profiles that are very useful for the reader.

The weaknesses of the presented work are already pointed out by the authors. Its limitations and possible biases are well defined and are shared with other similar studies on this topic.

Just point out additionally, the graphic simplicity in the presentation of the results which somewhat detracts the study and its effort.

Author Response

Dear esteemed reviewer,

We would like to express our sincere gratitude for your valuable time and expertise in reviewing our manuscript entitled "A Machine Learning approach unveils the relationships between Sickness Behavior and Interoception after vaccination: suggestions for psychometric indices of higher vulnerability". Your insights and recommendations have been immensely helpful in improving the quality and rigor of our research. In response to your constructive feedback, we have made the following modifications to address your concerns and enhance the overall quality of the paper.

______________________________________

We appreciate your acknowledgment of the study's strengths and the potential contributions.

As for your knowledge, all the changes required by the reviewers have been highlighted in yellow colour in the revised version of the manuscript.

Along with machine learning analysis, we have implemented the statistical analysis for the pre-post Sickness Behaviour comparison. For this purpose, we conducted a repeated measures ANOVA, resulting in a more consistent application of Cohen's d in line with the revised statistical approach. Regarding DASS-21 and PSQI variables, we initially employed ANOVA, incorporating the delta of pre-post scores on the Sickness Behaviour scale as a covariate when a significant pre-post effect was identified. This helped explore whether the observed effect could be explained by variations in Sickness Behaviour.

Additionally, we integrated Sickness Behaviour as a covariate in the inferential statistics approach. Conversely, for the Machine Learning approach, we dichotomized it. The ML approach, not reliant on a linear model, facilitates the identification of dichotomous rules. Consequently, predictors identified by ML may not necessarily align with those identified in a linear model estimated through multiple regression.

Once again, we appreciate the time and effort you have dedicated to reviewing our manuscript.

Reviewer 3 Report

Comments and Suggestions for Authors

The study reports some interesting findings and uses an innovative approach to differentiate groups of people with higher vs. levels of sickness behavior following the vaccination.

The topic is relevant and timely. Unfortunately, the organization of the manuscript and especially the description of the procedures lack some pieces of information which would make the results easier to comprehend and the conclusions easier to follow.

First of all – it should be stated immediately in the introductory parts of the manuscript that the participants did not fill the questionnaires in two separate occasions (before and after the vaccination) – rather, they  filled them in one sitting, reporting retrospectively about the symptoms before and the vaccination. The last sentence of the introduction explicitly mentions “baseline” emotional disposition, which implies measurements were taken before an intervention (in this case vaccination), which was not the case.  

Furthermore, it should be stated clearly in the methods section that the timelapse between the vaccination and responding the survey was one to three months after the vaccination – making some of these responses more biased than others (3 months is a lot of time, and while participants might be inclined to remember  symptoms occurring after the vaccination, the accuracy of reports of experiences taking place before the vaccination is highly questionable). Some descriptive data regarding the proportions of sample responding 1,2 or 3 months following the event might also be added. Ideally, one might include this variable as a covariate accounting for any systemic effects of time.

The instruments: please report reliability coefficients for questionnaires.

Also, please state specifically which scores for which subscales were used: for example, in the first reading one might assume that PSQI was treated as one – total score. However, later on in the results, it is revealed that sleep latency (which is only one facet of the questionnaire) was the most informative variable – and this is the first time in the manuscript that the authors mention treating these sub-scores separately (this is not just a semantics issue – it has potential implications for the interpretation of the results: if only one of 7 subscales showed predictive power, one has to wonder about the reliability of this finding).

The pre-post comparison: please explain the rationale for using a non-parametric test (Wilcoxon) instead of a parametric one (such as a paired samples t-test, ANOVA for repeated measures, or in this particular case an ANCOVA for repeated measures, with Sickness Behavior as a covariate..?)

The reasoning gets even more non-intuitive when the Cohen’s d is reported alongside with the non-parametric test (Cohen's d is derived from M and SD – while the effect size coefficients accompanying Wilcoxon  are derived from z-values and N – it is very unusual to see these two procedure used in the same analysis).

Related to this – the data in table 3 would be more informative if means and SDs were reported for DASS-21 and PSQI.

Also, the participants were categorized based of a 33/66 percentile cut-off on the SicknessQ score. However, categorizing continuous variables s rarely justified – could you please elaborate why you haven’t used this continuous variable as a covariate in the analyses, thus using the whole data range? I understand that for the second part of the study (the implementation of machine learning algorithm) it was necessary to have form groups, but for analyzing the T0/T1 differences, the continuous approach would be less biased and more informative.

Similarly, while I understand that the authors’ aim was to compare classification accuracy of different procedure, I would still prefer to see the complete output for the regression analysis (including the ponders for each individual predictor) at least for the most accurate analysis.

The discussion section is too short. It should emphasize the main findings but also elaborate more on their relevance. As it is now, the reader gets the impression that the authors were concerned mainly  with implementing a novel statistical tool, without clear reasoning behind it.  

Comments on the Quality of English Language

English is fine.

Author Response

Dear esteemed reviewer,

We would like to express our sincere gratitude for your valuable time and expertise in reviewing our manuscript entitled "A Machine Learning approach unveils the relationships between Sickness Behavior and Interoception after vaccination: suggestions for psychometric indices of higher vulnerability". Your insights and recommendations have been immensely helpful in improving the quality and rigor of our research. In response to your constructive feedback, we have made the following modifications to address your concerns and enhance the overall quality of the paper.

______________________________________

Regarding the first point, we have implemented the text as suggested, and it can be found at page 2 (lines 75-77).

“Taking advantage of the COVID-19 vaccination campaign, we collected data through an online survey to retrospectively investigate whether vaccine-induced SB could be modulated by Interoceptive Awareness alongside emotional disposition perceived before vaccination (at baseline).”

We appreciate the suggestion provided regarding the time interval between vaccination and questionnaire response. As you rightly pointed out, this time lapse between vaccination and survey responses can introduce some bias, particularly regarding the accuracy of pre-vaccination experiences. We included it within the "Limitation" section (page 9, lines 347-352).

“Thirdly, the methodological approach may be susceptible to a recall bias: the memory of the previous emotional state could be influenced by the emotional state experienced at the time of completing the questionnaire. Additionally, it is worth noting that the participants underwent the vaccination from one to three months before responding to the online survey. This can particularly impact the accuracy of pre-vaccination experiences.”

We have addressed your request by including the Cronbach's alpha values for each questionnaire used in our study (page 3, lines 104-115).

In the revised version all type subscales has been specified, and you can find them at page 4 (lines 185-189), as stated below:

“Additionally, based on the level of analysis, we progressed from studying potential predictors among the questionnaire total scores to evaluating specific questionnaire sub-scores and items contributing to scores that has been identified as predictors at the previous analysis level. For the classification, we only used the DASS-21, PSQI and MAIA questionnaire scores/sub-scores, as well as the corresponding items related to the baseline before vaccination, discarding any information from the week following vaccination.”

And at page 5 (lines 248-262), as stated below:

“As the final step, the classification was repeated focusing the analysis on the variables that JRIP indicated as exhibiting optimal efficiency in classifying the individuals within the two classes. Taking into account the eight JRIP rules the variables were: 1) DASS-21 subcomponents: Stress, Depression and Anxiety; 2) fever >38C; 3) systemic symptoms; 4) local symptoms; 5) concerns about adverse reactions; 6) MAIA Noticing, Attention Regulation and Body Listening; 7) sex; 8) age ≤ 50; 9) PSQI Sleep latency. In this particular instance, with the goal of identifying more specific aspects capable to predicting SB level after vaccination, the classification was performed using as independent variables the individual items contributing to all the aforementioned variables. Accordingly, we run the classifiers using as inputs: 1) age; 2) sex; 3) fever >38C; 4) systemic symptoms; 5) local symptoms; 6) concerns about adverse reaction; 7) MAIA items from 1 to 4 (MAIA-noticing); MAIA items from 11 to 17 (MAIA-attention regulation), MAIA items from 27 to 29 (MAIA-body listening); 8) PSQI items 2 and 5 (Sleep latency); 9) all items of DASS-21 (DASS-21 Stress, Depression, Anxiety). In this instance, JRIP yielded an accuracy of 75.37% with a simplified set of five rules:”.

Regarding the statistical approach, we initially took a conservative and broad approach by using non-parametric tests. However, upon considering your advice and noting that all variables follow gaussian distributions, we switched to parametric tests.

Therefore, as regards the pre-post Sickness Behavior comparison, we carried out a repeated measures ANOVA. Consequently, the reported Cohen's d is also more consistent with the statistical approach carried out.

As regards the variables of the DASS-21 and PSQI we performed the same ANOVA in the first instance, and where we identified a significant pre-post effect, we repeated the analysis incorporating the delta of pre-post scores on the Sickness Behavior scale as a covariate. This allowed us to investigate whether the pre-post effect could be explained by the variation in Sickness Behavior.

Therefore, following your suggestion, we integrated Sickness Behavior as a covariate in the inferential statistics approach. Conversely, for the Machine Learning approach, we dichotomized it. The ML approach doesn't rely on a linear model and enables the identification of dichotomous rules. Consequently, predictors identified by ML may not necessarily align with those identified in a linear model that could be estimated through multiple regression.

We have made some improvements to the discussion as per the suggestion (at page 8, lines 299-312; 322-337). Nevertheless, we would like to acknowledge that our current research methodology and the results obtained present limitations in our ability to draw definitive conclusions regarding the underlying mechanisms linking Sickness Behavior and interoceptive awareness. Our primary aim was to establish an initial association.

All changes have been highlighted in yellow color.

Round 2

Reviewer 3 Report

Comments and Suggestions for Authors

Thank you for addressing some of my comments.

I am still puzzled by some of your answers, but as they do not appear in the manuscript itself, it does not matter.

(For egzample, I do not understand why you think that adding the Cronbach alpha values addresses the problem of biased recall ? Specifically, to my comment that "....it is worth noting that the participants underwent the vaccination from one to three months before responding to the online survey. This can particularly impact the accuracy of pre-vaccination experiences.” you responded with "We have addressed your request by including the Cronbach's alpha values for each questionnaire used in our study (page 3, lines 104-115).

Comments on the Quality of English Language

Minor editing needed.